# Training Private Models That Know What They Don't Know

**Stephan Rabanser**[1,2]*, **Anvith Thudi**[1,2], **Abhradeep Thakurta**[3],
**Krishnamurthy (Dj) Dvijotham**[3], **Nicolas Papernot**[1,2]
[1]University of Toronto, [2]Vector Institute, [3]Google DeepMind

## Abstract

Training reliable deep learning models which avoid making overconfident but incorrect predictions is a longstanding challenge. This challenge is further exacerbated when learning has to be differentially private: protection provided to sensitive data comes at the price of injecting additional randomness into the learning process. In this work, we conduct a thorough empirical investigation of selective classifiers—that can abstain under uncertainty—under a differential privacy constraint. We find that some popular selective prediction approaches are ineffective in a differentially private setting because they increase the risk of privacy leakage. At the same time, we identify that a recent approach that only uses checkpoints produced by an off-the-shelf private learning algorithm stands out as particularly suitable under DP. Further, we show that differential privacy does not just harm utility but also degrades selective classification performance. To analyze this effect across privacy levels, we propose a novel evaluation mechanism which isolates selective prediction performance across model utility levels at full coverage. Our experimental results show that recovering the performance level attainable by non-private models is possible but comes at a considerable coverage cost as the privacy budget decreases.

## 1  Introduction

State of the art machine learning (ML) models are gaining rapid adoption in high-stakes application scenarios such as healthcare [Challen et al., 2019, Mozannar and Sontag, 2020], finance [Vijh et al., 2020], self-driving [Ghodsi et al., 2021], and law [Vieira et al., 2021]. However, major challenges remain to be addressed to ensure trustworthy usage of these models. One major concern in sensitive applications is to protect the privacy of the individuals whose data an ML model was trained on. To prevent privacy attacks on ML models, $(\varepsilon, \delta)$ differential privacy (DP) [Dwork et al., 2014] has emerged as the de-facto standard with widespread usage in both academic and industrial applications.

While the introduction of DP successfully protects against privacy attacks, it also limits model utility in practice. For instance, the canonical algorithm for training models with DP, DP-SGD [Abadi et al., 2016], clips the per-sample gradient norm and adds carefully calibrated noise. These training-time adaptations frequently lead to degradation in predictive performance in practice. This is especially a problem for datasets containing underrepresented subgroups whose accuracy has been shown to degrade with stronger DP guarantees [Bagdasaryan et al., 2019]. Faced with this challenge, it is therefore of vital importance to detect samples on which a DP model would predict incorrectly on.

One popular technique used to detect inputs that the model would misclassify with high probability is given by the selective classification (SC) framework [Geifman and El-Yaniv, 2017]: by relying on a model's uncertainty in the correct prediction for an incoming data point, the point is either accepted with the underlying model's prediction, or rejected and potentially flagged for additional

---

*Correspondence to: `stephan@cs.toronto.edu`

37th Conference on Neural Information Processing Systems (NeurIPS 2023).

downstream evaluation. Hence, SC introduces a trade-off between coverage over the test set (i.e., the goal of accepting as many data points as possible) and predictive performance on the accepted points (i.e., the goal of making as few mistakes as possible). Although identification of misclassified points seems of particular importance under differential privacy, the application of selective prediction to private models is under-explored to date. While lots of past works have studied privacy or selective prediction in isolation, best practices for selective prediction under differential privacy have not been established yet. In this work, we are the first to answer the question of whether selective classification can be used to recover the accuracy lost by applying a DP algorithm (at the expense of data coverage).

To analyze the interplay between differential privacy and selective classification, we first show that not all approaches towards selective classification are easily applicable under a differential privacy constraint. In particular, approaches relying on multiple passes over the entire dataset to obtain the full SP performance profile [Geifman and El-Yaniv, 2019, Lakshminarayanan et al., 2017] suffer significantly under differential privacy. This is due to the fact that the worst-case privacy leakage increases with each analysis made of the dataset which means that each training run forces more privacy budget to be expended. On the other hand, our analysis shows that an SC approach based on harnessing intermediate model checkpoints [Rabanser et al., 2022] yields the most competitive results. Notably, this approach performs especially well under stringent privacy constraints ($\varepsilon = 1$).

Next, we find that differential privacy has a more direct negative impact on selective classification that goes beyond the characteristic drop in utility. Based on a simple synthetic experiment, we observe a clear correlation between differential privacy strength and wrongful overconfidence. In particular, even when multiple private models trained with different $\varepsilon$ levels offer the same utility on the underlying prediction task, we show that stronger levels of DP lead to significantly decreased confidence in the correct class. This reduction effectively prevents performant selective classification.

Motivated by this observation, we realize a need to disentangle selective classification from model utility; SC ability and accuracy should be considered independently. To that end, we show that the evaluation metric typically used for non-private selective classification is inappropriate for comparing SC approaches across multiple privacy levels. The in-applicability of the current method stems from the fact that full-coverage accuracy alignment is required across experiments. In particular, accuracy-aligning multiple DP models via early-stopping frequently leads to a more stringent than targeted privacy level, rendering the intended comparison impossible. Due to this limitation of previous evaluation schemes, we propose a novel metric which isolates selective classification performance from losses that come from accuracy degradation of the classifier overall (as frequently caused by DP). The performance metric we propose to tackle this problem computes an upper bound on the accuracy/coverage trade-off in a model-dependent fashion and measures the discrepancy between the actual achieved trade-off to this bound. Using this score, we determine that, based on a comprehensive experimental study, differential privacy does indeed harm selective prediction beyond a loss in utility.

We summarize our key contributions below:

1. We provide a first analysis on the interplay between selective classification and differential privacy. As a result, we identify an existing SC method as particularly suitable under DP and present an illustrative example showcasing an inherent tension between privacy and selective prediction.

2. We unearth a critical failure mode of the canonical SC performance metric preventing its off-the-shelf usage under DP experimentation. We remedy this issue by introducing a novel accuracy-dependent selective classification score which enables us to compare selective classification performance across DP levels without explicit accuracy-alignment.

3. We conduct a thorough empirical evaluation across multiple selective classification techniques and privacy levels. As part of this study, we confirm that selective classification performance degrades with stronger privacy and further find that recovering utility can come at a considerable coverage cost under strong privacy requirements.

## 2 Background

### 2.1 Differential Privacy with DP-SGD

*Differential Privacy (DP)* [Dwork et al., 2014] is a popular technique which allows us to reason about the amount of privacy leakage from individual data points. Training a model with differential privacy

ensures that the information content that can be acquired from individual data points is bounded while patterns present across multiple data points can still be extracted. Formally, a randomized algorithm $\mathcal{M}$ satisfies $(\varepsilon, \delta)$ differential privacy, if for any two datasets $D, D' \subseteq \mathcal{D}$ that differ in any one record and any set of outputs $S$ the following inequality holds:

$$\mathbb{P}\left[\mathcal{M}(D) \in S\right] \leq e^{\varepsilon} \mathbb{P}\left[\mathcal{M}(D') \in S\right] + \delta \tag{1}$$

The above DP bound is governed by two parameters: $\varepsilon \in \mathbb{R}_+$ which specifies the privacy level, and $\delta \in [0, 1]$ which allows for a small violation of the bound.

**DP-SGD.** A canonical way of incorporating differential privacy in deep learning is via *Differentially Private Stochastic Gradient Descent (DP-SGD)* [Bassily et al., 2014, Abadi et al., 2016]. We describe the DP-SGD algorithm in detail in Algorithm 2 in the Appendix. The main adaptations needed to incorporate DP into SGD are: (i) *per-sample gradient computation*, which allows us to limit the per-point privacy leakage in the next two steps; (ii) *gradient clipping*, which bounds the sensitivity (i.e., the maximal degree of change in the outputs of the DP mechanism); and (iii) *noise addition*, which introduces Gaussian noise proportional to the intensity of gradient clipping.

## 2.2 Selective Classification

**Supervised classification.** Our work considers the supervised classification setup: We assume access to a dataset $D = \{(\boldsymbol{x}_n, y_n)\}_{n=1}^{N}$ consisting of data points $(\boldsymbol{x}, y)$ with $\boldsymbol{x} \in \mathcal{X} \subseteq \mathbb{R}^d$ and $y \in \mathcal{Y} = \{1, \ldots, C\}$. All data points $(\boldsymbol{x}, y)$ are sampled independently and identically distributed from the underlying distribution $p$ defined over $\mathcal{X} \times \mathcal{Y}$. The goal is to learn a prediction function $f : \mathcal{X} \to \mathcal{Y}$ which minimizes the classification risk with respect to the underlying data distribution $p$ as measured by a loss function $\ell : \mathcal{Y} \times \mathcal{Y} \to \mathbb{R}$.

**Selective classification.** Selective classification augments the supervised classification setup by introducing a rejection class $\perp$ via a *gating mechanism* [El-Yaniv and Wiener, 2010]. This mechanism produces a class label from the underlying classifier if the mechanism is confident that the prediction is correct and abstains otherwise. Note that this mechanism is often directly informed by the underlying classifier $f$ and we make this dependence explicit in our notation. More formally, the gating mechanism introduces a selection function $g : \mathcal{X} \times (\mathcal{X} \to \mathcal{Y}) \to \mathbb{R}$ which determines if a model should predict on a data point $\boldsymbol{x}$. If the output of $g(\boldsymbol{x}, f)$ undercuts a given threshold $\tau$, we return $f(\boldsymbol{x})$, otherwise we abstain with decision $\perp$. The joint predictive model is therefore given by:

$$(f, g)(\boldsymbol{x}) = \begin{cases} f(\boldsymbol{x}) & g(\boldsymbol{x}, f) \leq \tau \\ \perp & \text{otherwise.} \end{cases} \tag{2}$$

**Evaluating SC performance.** The performance of a selective classifier $(f, g)$ is based on two metrics: the *coverage* of $(f, g)$ (corresponding to the fraction of points to predict on) and the *accuracy* of $(f, g)$ on accepted points. Note that there exists a tension between these two performance quantities: naively increasing coverage will lead to lower accuracy while an increase in accuracy will lead to lower coverage. Successful SC methods try to jointly maximize both metrics.

$$\text{cov}_\tau(f, g) = \frac{|\{\boldsymbol{x} : g(\boldsymbol{x}, f) \leq \tau\}|}{|D|} \qquad \text{acc}_\tau(f, g) = \frac{|\{\boldsymbol{x} : f(\boldsymbol{x}) = y, g(\boldsymbol{x}, f) \leq \tau\}|}{|\{\boldsymbol{x} : g(\boldsymbol{x}, f) \leq \tau\}|} \tag{3}$$

To characterize the full performance profile of a selective classifier $(f, g)$, we consider the *selective accuracy* at coverage level $c$, formally $\text{acc}_c(f, g)$, over the full coverage spectrum by computing an area-under-the-curve (AUC) score $s_{\text{AUC}}$ as follows:

$$s_{\text{AUC}}(f, g) = \int_0^1 \text{acc}_c(f, g) dc \qquad \text{acc}_c(f, g) = \text{acc}_\tau(f, g) \quad \text{for } \tau \text{ s.t. } \text{cov}_\tau(f, g) = c \tag{4}$$

Each existing selective classification algorithm proposes a $g$ that tries to maximize this metric. We describe popular approaches below and include additional details in Appendix A.2.

**Prediction confidence.** The traditional baseline methods for selective prediction is the *Softmax Response* (SR) method [Hendrycks and Gimpel, 2016, Geifman and El-Yaniv, 2017]. This method uses the confidence of the final prediction model $f$ as the selection score. To reduce overconfident predictions yielded by this method, confidence calibration [Guo et al., 2017] has been proposed.

**Ensembling.**    To further improve calibration and to reduce variance, ensemble methods have been proposed which combine the information content of $M$ models into a single final model. The canonical instance of this approach for deep learning based models, *Deep Ensembles* (DE) [Lakshminarayanan et al., 2017], trains multiple models from scratch with varying initializations using a proper scoring rule and adversarial training. Then, after averaging the predictions made by all models, the softmax response (SR) mechanism is applied. To overcome the computational cost of estimating multiple models from scratch, *Monte-Carlo Dropout* (MCDO) [Gal and Ghahramani, 2016] allows for bootstrapping of model uncertainty of a dropout-equipped model at test time. Another ensembling approach that has recently demonstrated state-of-the-art selective classification performance is based on monitoring model evolution during the training process. To that end, *Selective Classification Training Dynamics* (SCTD) [Rabanser et al., 2022] records intermediate models produced during training and computes a disagreement score of intermediate predictions with the final prediction.

**Architecture & loss function modifications.**    A variety of SC methods have been proposed that leverage explicit architecture and loss function adaptations. For example, *SelectiveNet* (SN) [Geifman and El-Yaniv, 2019] modifies the model architecture to jointly optimize $(f, g)$ while targeting the model at a desired coverage level $c_{\text{target}}$. Alternatively, prior works like *Deep Gamblers* [Liu et al., 2019] and *Self-Adaptive Training* [Huang et al., 2020] have considered explicitly modeling the abstention class $\perp$ and adapting the optimization process to provide a learning signal for this class. For instance, *Self-Adaptive Training* (SAT) incorporates information obtained during the training process into the optimization itself by computing and monitoring an exponential moving average of training point predictions over the training process. Introduced to yield better uncertainty estimates, Liu et al. [2020] employs weight normalization and replaces the output layer of a neural network with a Spectral-Normalized Neural Gaussian Process to improve data manifold characterization.

**Uncertainty for DP models.**    An initial connection between differential privacy and uncertainty quantification is explored in Williams and McSherry [2010], which shows that probabilistic inference can improve accuracy and measure uncertainty on top of differentially private models. By relying on intermediate model predictions, Shejwalkar et al. [2022] has proposed a mechanism to quantify the uncertainty that DP noise adds to the outputs of ML algorithms (without any additional privacy cost).

## 3   The Interplay Between Selective Classification and Differential Privacy

We now examine the interplay between these approaches to selective classification and training algorithms constrained to learn with differential privacy guarantees. We first introduce a candidate selective classification approach which leverages intermediate predictions from models obtained during training. Then, we study how current selective classification approaches impact DP guarantees, as each access they make to training data increases the associated privacy budget. Next, we investigate how in turn DP affects selective classification performance. Indeed, DP alters how optimizers converge to a solution and as such DP can impact the performance of SC techniques. Last, we discuss how selective classification performance can be fairly compared across a range of privacy levels.

---

**Algorithm 1:** SCTD [Rabanser et al., 2022]

---

**Require:** Checkpointed model sequence $\{f_1, \ldots, f_T\}$, query point $\boldsymbol{x}$, weighting parameter $k \in [0, \infty)$.

1: Compute prediction of last model: $L \leftarrow f_T(\boldsymbol{x})$
2: Compute disagreement and weighting of intermediate predictions:
3: **for** $t \in [T]$ **do**
4:     **if** $f_t(\boldsymbol{x}) = L$ **then** $a_t \leftarrow 0$ **else** $a_t \leftarrow 1$
5:     $v_t \leftarrow (\frac{t}{T})^k$
6: **end for**
7: Compute sum score: $s_{\text{sum}} \leftarrow \sum_t v_t a_t$
8: **if** $s_{\text{sum}} \leq \tau$ **then** accept $f(\boldsymbol{x}) = L$ **else** reject with $f(\boldsymbol{x}) = \perp$

---

### 3.1   Performative Private Selective Classification via Training Dynamics Ensembles

As a classical technique from statistics, ensembling methods are often employed for confidence interval estimation [Karwa and Vadhan, 2017, Ferrando et al., 2022], uncertainty quantification [Lakshminarayanan et al., 2017], and selective prediction [Zaoui et al., 2020]. Under a differential privacy constraint, although for simpler tasks like mean estimation ensembling methods have been proven to be effective [Brawner and Honaker, 2018, Covington et al., 2021, Evans et al., 2019], in this paper

we demonstrate that they are fairly ineffective for selective classification. This is primarily due to the increased privacy cost due to (advanced sequential) composition [Dwork et al., 2006].

In light of this challenge, we expect one recent method in particular to perform well in a private learning setting: selective classification via training dynamics (SCTD) [Rabanser et al., 2022] (details described in Algorithm 1). For a given test-time point, SCTD analyzes the disagreement with the final predicted label over intermediate models obtained during training. Data points with high disagreement across this training-time ensemble are deemed anomalous and rejected. Most importantly, while SCTD also constructs an ensemble, only a single training run is used to obtain this ensemble. As a result of the post-processing property of DP, the original $(\varepsilon, \delta)$-DP guarantee can be maintained. To further motivate the effectiveness of relying on intermediate checkpoints, Shejwalkar et al. [2022] has shown in adjacent work that intermediate predictions can improve the predictive accuracy of DP training, yielding new state-of-the-art results under DP.

### 3.2 How Do Other Selective Classification Approaches Affect Privacy Guarantees?

We can extend the previous analysis based on post-processing and composition to other SC techniques. This allows us to group selective classification techniques into three classes: (i) directly optimized; (ii) post-processing; and (iii) advanced sequential composition. For directly optimized and post-processing approaches, we can obtain selective classification for free. On the other hand, composition-based approaches either require an increased privacy budget or suffer from decreased utility.

**Direct optimization.** Many selective classification approaches directly modify the loss function and optimization is performed w.r.t. this adapted loss function. As DP-SGD is loss function and architecture agnostic $(\varepsilon, \delta)$ guarantees hold automatically for SC methods that only change the loss function. *Applicable to:* Softmax Response (SR), Deep Gamblers (DG), Self-Adaptive Training (SAT).

**Post-processing.** If a function $\phi(x)$ satisfies $(\varepsilon, \delta)$-DP, then for any deterministic or random function $\psi(\cdot)$, the application of $\psi$ on $\phi$, formally $\psi \circ \phi(x)$, still satisfies $(\varepsilon, \delta)$-DP [Dwork et al., 2006]. *Applicable to:* Monte-Carlo Dropout (MCDO), Selective Classification Training Dynamics (SCTD).

**Advanced sequential composition.** If in a set of aggregated functions $\{\phi_1(x), \ldots, \phi_M(x)\}$ each $\phi_i(x)$ satisfies $(\varepsilon, \delta)$-DP, then releasing the set of all outputs $\psi(x) = (\phi_1(x), \ldots, \phi_M(x))$ satisfies $\approx (\sqrt{M}\varepsilon, M\delta)$-DP [Dwork et al., 2006]. To maintain $(\varepsilon, \delta)$-DP over the aggregated output, each function needs to satisfy $\approx (\frac{\varepsilon}{\sqrt{M}}, \frac{\delta}{M})$-DP. *Applicable to*: Deep Ensembles (DE), SelectiveNet (SN).

### 3.3 How Does Differential Privacy Affect Selective Classification Performance?

After having examined how current selective classification techniques influence differential privacy guarantees, this subsection considers the opposite effect: what is the impact of differential privacy on selective classification? As we will see by means of an intuitive example, we expect that differential privacy impacts selective classification beyond a loss in utility.

To showcase this, we present a synthetic logistic regression example with the softmax response SC mechanism. We generate data from a mixture of two two-dimensional Gaussians with heavy class-imbalance. Concretely, we generate samples for a majority class $\{(\boldsymbol{x}_i, 1)\}_{i=1}^{1000}$ with each $\boldsymbol{x}_i \sim \mathcal{N}(\boldsymbol{0}, \boldsymbol{I})$ and a single outlier point form a minority class $(\boldsymbol{x}^*, -1)$ with $\boldsymbol{x}^* \sim \mathcal{N}([10 \quad 0]^\top, \boldsymbol{I})$. We then train multiple differentially private mechanisms with $\varepsilon \in \{\infty, 7, 3, 1\}$ on this dataset and evaluate all models on a test set produced using the same data-generating process.

We show the outcome of this set of experiments in Figure 1 where we plot the test set and the decision boundary of each model across $\varepsilon$ levels. For $\varepsilon = \infty$, the model successfully discriminates the majority and the minority class. However, the decision boundary is heavily influenced by a single data point, namely the outlier $(\boldsymbol{x}^*, -1)$. Note that, even though all models with $\varepsilon \in \{7, 3, 1\}$ misclassify the outlier point (i.e., their utility is aligned), the changing decision boundary also increases the model's confidence in predicting the incorrect class for the outlier. Hence, even at an aligned utility level, the softmax response approach for SC performs increasingly worse on the outlier point under stronger DP constraints. We provide additional class-imbalanced results on realistic datasets in Appendix B.2.

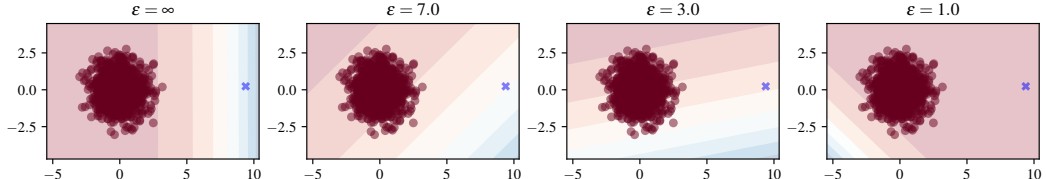

Figure 1: **Synthetic example of privacy impacts on selective classification**. Shaded regions show softmax response confidence in the red and blue class, respectively. As $\varepsilon$ decreases, the blue outlier point is increasingly protected by DP. Importantly, not only does DP induce a misclassification on the outlier point, but stronger levels of DP further increase confidence in the wrong prediction. This overconfidence in the incorrect class prevents selective classification.

### 3.4 How Should We Evaluate Selective Classification Under Differential Privacy?

As we have now established, we expect differential privacy to harm selective classification performance. In light of this insight, we are now investigating whether the standard evaluation scheme for SC is directly applicable to the differentially private models.

The default approach of using the SC performance metric introduced in Equation 4 free from its bias towards accuracy is to align different SC approaches/models at the same accuracy. However, accuracy-aligning can have unintended consequences on SC performance, especially under DP. If we are considering SC performance across models trained for multiple $\varepsilon$ levels, accuracy-alignment would require lowering the performance of the less private models. One seemingly suitable approach to do this is by early-stopping model training at a desired accuracy level. However, early-stopping a DP training algorithm in fact gives a model with greater privacy than the privacy level targeted by the optimization process. This is a direct consequence of privacy loss accumulation via privacy accounting during training, and so training for less leads to expending less privacy budget. Hence, instead of comparing models with different privacy guarantees at the same baseline utility, early-stopping yields models with potentially very similar privacy guarantees.

To address this limitation, we propose to measure SC performance by how close the observed trade-off is to an upper bound on the achievable SC performance at a given baseline utility level (at 100% coverage). To compute this metric, we first derive an upper bound on selective classification performance (as a function of coverage) for a given baseline accuracy. We obtain this characterization by identifying the optimal acceptance ordering under a particular full-coverage accuracy constraint.

**Definition 3.1.** *The upper bound on the selective classification performance for a fixed full-coverage accuracy $a_{full} \in [0, 1]$ and a variable coverage level $c \in [0, 1]$ is given by*

$$\overline{acc}(a_{full}, c) = \begin{cases} 1 & 0 < c \leq a_{full} \\ \frac{a_{full}}{c} & a_{full} < c < 1 \end{cases}. \tag{5}$$

Measuring the area enclosed between the bound $\overline{acc}(a_{full}, c)$ and an SC method's achieved accuracy/coverage trade-off $acc_c(f, g)$ yields our accuracy-normalized selective classification score.

**Definition 3.2.** *The accuracy-normalized selective classification score $s_{a_{full}}(f, g)$ for a selective classifier $(f, g)$ with full-coverage accuracy $a_{full}$ is given by*

$$s_{a_{full}}(f, g) = \int_0^1 (\overline{acc}(a_{full}, c) - acc_c(f, g))dc \approx \sum_c (\overline{acc}(a_{full}, c) - acc_c(f, g)). \tag{6}$$

We provide additional intuition for both the proposed upper bound and our score as part of Figure 2.

**Bound justification.** To understand the upper bound given in Equation 5, note that a full-coverage accuracy level of $a_{full}$ means that a fraction $a_{full}$ of points are correctly classified. An ideal selective classification method for this model would, by increasing the coverage $c$ from 0 to 1, first accept all points that the model classifies correctly: this gives the optimal selective classification accuracy for coverage $c \leq a_{full}$ of 100%. For any coverage level $c > a_{full}$, note that the accuracy at the coverage is

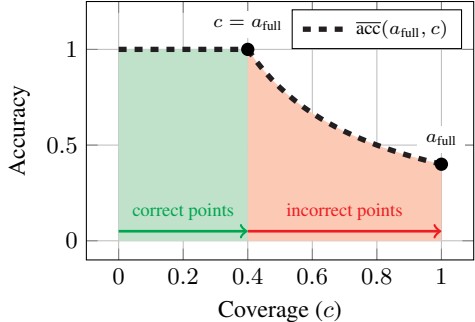
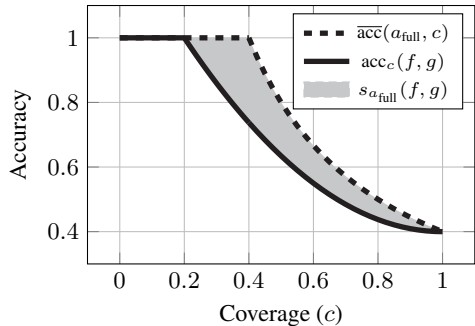

(a) *Upper bound* $\overline{acc}(a_{full}, c)$ *on the selective classification performance conditional on* $a_{full}$. As coverage $c$ increases from $0 \to 1$, an optimal selective classifier accepts all correct points first (with a consistent accuracy of 1 until $c = a_{full}$) and then spreads the accuracy decrease $\frac{a_{full}}{c}$ equally over the remaining coverage spectrum, leading to a convex drop as $c \to 1$.

(b) *Accuracy-normalizes score for SC performance.* The accuracy-normalized score for selective classification $s_{a_{full}}(f, g)$ corresponds to the area enclosed between the upper bound $\overline{acc}(a_{full}, c)$ and the empirical accuracy/coverage trade-off $acc_c(f, g)$. Good selective classifiers should achieve a low score ($s_{a_{full}}(f, g) \approx 0$), indicating closeness to the bound.

Figure 2: **Selective classification upper bound and accuracy-normalized score visualization**. We present an example of a selective classifier with full-coverage accuracy $a_{full} = 0.4$ and show how (a) the corresponding upper bound; and how (b) the accuracy-normalized score can be obtained.

the number of correct points accepted divided by the total number of points accepted. The largest value this can take is by maximizing the numerator, and the maximum value for this (as a fraction of all points) is $a_{full}$. Plugging this into the previous statement for accuracy at coverage $c$, we have the best selective accuracy at a coverage $c$ is $\frac{a_{full}}{c}$. This derives the upper bound stated above as a function of coverage. We remark that this bound is in fact achievable if a selective classifier separates correct and incorrect points perfectly, i.e., it accepts all correct points first and then accepts and increasing amount of incorrect points as we increase the coverage of the selective classification method. See Appendix B.3 for an experiment that matches the bound exactly.

# 4  Experiments

We now present a detailed experimental evaluation of the interplay between differential privacy and selective classification. All of our experiments are implemented using PyTorch [Paszke et al., 2019] and Opacus [Yousefpour et al., 2021]. We publish our full code-base at the following URL: `https://github.com/cleverhans-lab/selective-classification`.

**Setup.**  Our evaluation is based on the following experimental panel. In particular, we conduct experiments for each dataset/SC-method/$\varepsilon$ combination:

- **Datasets**: FashionMNIST [Xiao et al., 2017], CIFAR-10 [Krizhevsky et al., 2009], SVHN [Netzer et al., 2011], GTSRB [Houben et al., 2013].

- **Selective Classification Methods**: Softmax Response (`SR`) [Geifman and El-Yaniv, 2017], SelectiveNet (`SN`) [Geifman and El-Yaniv, 2019], Self-Adaptive Training (`SAT`) [Huang et al., 2020], Monte-Carlo Dropout (`MCDO`) [Gal and Ghahramani, 2016], Deep Ensembles (`DE`) [Lakshminarayanan et al., 2017], Selective Classification Training Dynamics (`SCTD`) [Rabanser et al., 2022].

- **Privacy levels**: $\varepsilon \in \{\infty, 7, 3, 1\}$.

Based on recent results from Feng et al. [2023], we (i) apply additional entropy regularization to all methods; and (ii) derive the selective classification scores for `SN` and `SAT` by applying Softmax Response (`SR`) to the underlying classifier (instead of relying on model-specific abstention mechanisms). Across all private experiments, we set $\delta = 10^{-(N+1)}$ for datasets consisting of training samples on the order of $10^N$. For each combination of SC method, dataset, and privacy level, we train a model following the ResNet-18 [He et al., 2016] model architecture using either the SGD optimizer (for $\varepsilon = \infty$) or the DP-SGD optimizer (for all $\varepsilon < \infty$). All models are trained for 200 epochs using a

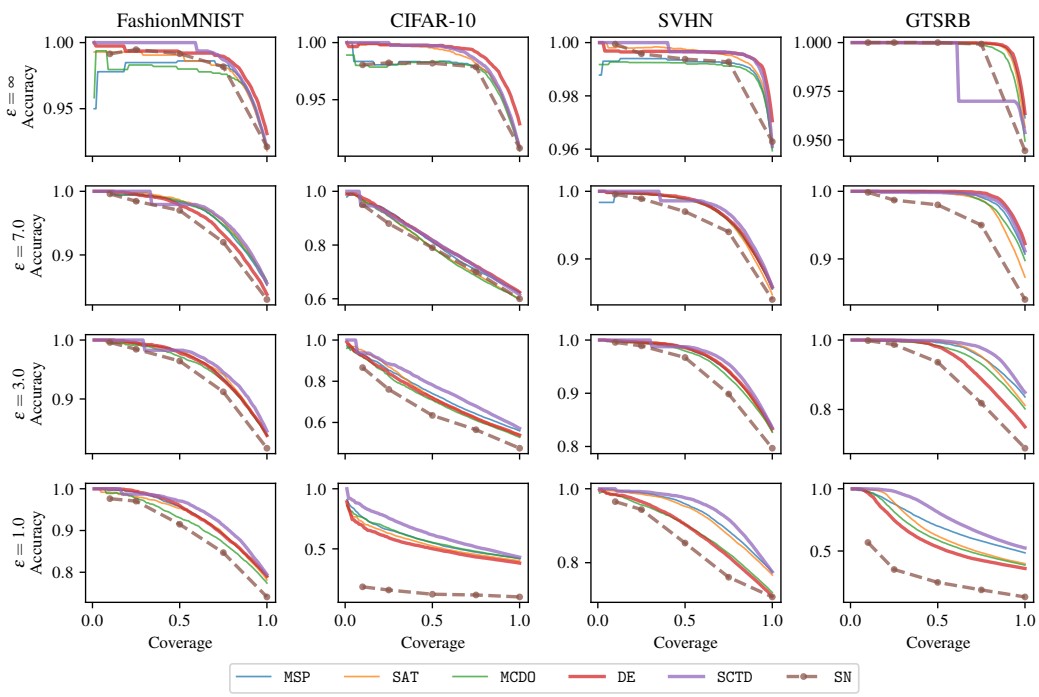

Figure 3: **Accuracy-coverage trade-off across datasets & $\varepsilon$ levels**. We highlight noteworthy results for SCTD, DE, and SN with bold lines and further show that SN only provides results at coarse-grained coverage levels. SCTD performs best while DE and SN performance is compromised at any $\varepsilon < \infty$.

Table 1: **Coverage required for non-private full-coverage accuracy.** We observe that differential privacy considerably lowers the allowable coverage to achieve a utility level consistent with the non-private model. Across our experiments, SCTD yields the highest coverage across $\varepsilon$ levels.

| | FashionMNIST | | | CIFAR-10 | | | SVHN | | | GTSRB | | |
|---|---|---|---|---|---|---|---|---|---|---|---|---|
| | $\varepsilon = 7$ | $\varepsilon = 3$ | $\varepsilon = 1$ | $\varepsilon = 7$ | $\varepsilon = 3$ | $\varepsilon = 1$ | $\varepsilon = 7$ | $\varepsilon = 3$ | $\varepsilon = 1$ | $\varepsilon = 7$ | $\varepsilon = 3$ | $\varepsilon = 1$ |
| MSP | 0.83 (±0.01) | 0.80 (±0.01) | 0.65 (±0.03) | **0.29 (±0.02)** | 0.14 (±0.04) | 0.00 (±0.00) | 0.74 (±0.00) | 0.67 (±0.01) | 0.49 (±0.02) | 0.90 (±0.01) | 0.71 (±0.03) | 0.13 (±0.00) |
| SAT | **0.86 (±0.00)** | 0.81 (±0.01) | 0.67 (±0.02) | 0.25 (±0.01) | **0.19 (±0.02)** | 0.00 (±0.00) | 0.72 (±0.00) | 0.67 (±0.01) | 0.45 (±0.02) | 0.86 (±0.00) | 0.74 (±0.00) | 0.20 (±0.03) |
| MCDO | **0.84 (±0.02)** | 0.79 (±0.00) | 0.56 (±0.02) | 0.25 (±0.01) | 0.12 (±0.02) | 0.00 (±0.00) | 0.74 (±0.00) | 0.64 (±0.00) | 0.23 (±0.03) | 0.90 (±0.01) | 0.69 (±0.01) | 0.14 (±0.01) |
| DE | 0.75 (±0.00) | 0.75 (±0.01) | 0.61 (±0.01) | 0.22 (±0.01) | 0.09 (±0.00) | 0.00 (±0.00) | 0.69 (±0.01) | 0.62 (±0.01) | 0.22 (±0.00) | **0.93 (±0.00)** | 0.57 (±0.08) | 0.10 (±0.04) |
| SCTD | **0.86 (±0.01)** | **0.84 (±0.02)** | **0.73 (±0.01)** | 0.26 (±0.03) | **0.20 (±0.03)** | **0.04 (±0.04)** | **0.78 (±0.01)** | **0.72 (±0.00)** | **0.59 (±0.02)** | **0.93 (±0.01)** | **0.83 (±0.03)** | **0.30 (±0.02)** |

mini-batch size of 128. For all DP training runs, we set the clipping norm to $c = 10$ and choose the noise multiplier adaptively to reach the overall privacy budget at the end of training. Note that, since we want to consider a consistent value of $\varepsilon$ across all SC methods, we restrict the overall $\varepsilon$ in both SN and DE. As explained in Section 3.2, this effectively enforces a reduction in $\varepsilon$ for each individual training run due to sequential composition. Concretely, we train DE with 5 ensemble members and SN with target coverage levels $c_{\text{target}} \in \{0.1, 0.25, 0.5, 0.75, 1.0\}$. This leads to a reduction of $\approx \frac{\varepsilon}{\sqrt{5}}$ for each trained sub-model in both cases. We account for the precise reduction in the privacy parameter by *increasing by the number steps of DP-SGD* as measured by a Rényi DP accountant [Mironov, 2017, Yousefpour et al., 2021]. All experiments are repeated over 5 random seeds to determine the statistical significance of our findings. We document additional hyper-parameters in Appendix B.1.

**Evaluation of selective classification techniques across privacy levels.** We report our main results in Figure 3 where we display the accuracy-coverage trade-off for our full experimental panel. Overall, we observe that Selective Classification Training Dynamics (SCTD) delivers the best accuracy-coverage trade-offs. While this is consistent with non-private [Rabanser et al., 2022], we find that SCTD delivers especially strong performance at lower $\varepsilon$ levels. This suggests that leveraging training dynamics information is useful for performative selective classification under DP. As expected, based off of the discussion in Section 3.2, Figure 3 also shows that the performance of both Deep Ensembles (DE) and SelectiveNet (SN) suffers under DP, showing increasing utility degradation as $\varepsilon$ decreases.

Table 2: **Accuracy-normalized selective classification performance.** Across our panel of experiments, we find that decreasing $\varepsilon$ leads to a worse (i.e., higher) score, meaning that as $\varepsilon$ decreases the selective classification approaches all move away from the upper bound.

|  | FashionMNIST | | | | CIFAR-10 | | | |
|---|---|---|---|---|---|---|---|---|
|  | $\epsilon = \infty$ | $\epsilon = 7$ | $\epsilon = 3$ | $\epsilon = 1$ | $\epsilon = \infty$ | $\epsilon = 7$ | $\epsilon = 3$ | $\epsilon = 1$ |
| MSP | 0.019 (±0.000) | 0.023 (±0.000) | 0.027 (±0.002) | 0.041 (±0.001) | 0.019 (±0.000) | 0.105 (±0.002) | 0.133 (±0.002) | 0.205 (±0.001) |
| SAT | 0.014 (±0.000) | **0.020 (±0.001)** | 0.026 (±0.002) | 0.043 (±0.002) | 0.010 (±0.000) | 0.107 (±0.000) | 0.128 (±0.000) | 0.214 (±0.002) |
| MCDO | 0.020 (±0.002) | 0.023 (±0.001) | 0.030 (±0.003) | 0.053 (±0.001) | 0.021 (±0.001) | 0.110 (±0.000) | 0.142 (±0.000) | 0.201 (±0.000) |
| DE | **0.010 (±0.003)** | 0.027 (±0.002) | 0.027 (±0.002) | 0.039 (±0.000) | **0.007 (±0.001)** | **0.099 (±0.002)** | 0.138 (±0.000) | 0.222 (±0.000) |
| NNTD | **0.007 (±0.001)** | **0.021 (±0.001)** | **0.023 (±0.003)** | **0.032 (±0.002)** | 0.009 (±0.002) | **0.098 (±0.001)** | **0.107 (±0.001)** | **0.152 (±0.001)** |
| SN | **0.008 (±0.002)** | 0.058 (±0.001) | 0.056 (±0.001) | 0.064 (±0.002) | 0.015 (±0.000) | 0.155 (±0.003) | 0.154 (±0.002) | 0.173 (±0.001) |

|  | SVHN | | | | GTSRB | | | |
|---|---|---|---|---|---|---|---|---|
| MSP | 0.008 (±0.001) | 0.020 (±0.001) | 0.024 (±0.001) | 0.040 (±0.001) | **0.001 (±0.001)** | 0.006 (±0.002) | 0.017 (±0.000) | 0.109 (±0.002) |
| SAT | **0.004 (±0.000)** | 0.019 (±0.001) | **0.021 (±0.002)** | 0.044 (±0.002) | **0.001 (±0.001)** | 0.008 (±0.000) | 0.014 (±0.000) | 0.089 (±0.000) |
| MCDO | 0.009 (±0.000) | 0.019 (±0.001) | 0.027 (±0.002) | 0.069 (±0.001) | 0.002 (±0.001) | 0.007 (±0.001) | 0.023 (±0.001) | 0.110 (±0.001) |
| DE | **0.004 (±0.001)** | 0.018 (±0.001) | 0.022 (±0.002) | 0.067 (±0.003) | **0.001 (±0.000)** | **0.003 (±0.002)** | 0.027 (±0.004) | 0.127 (±0.002) |
| NNTD | **0.003 (±0.001)** | **0.016 (±0.001)** | **0.018 (±0.002)** | **0.027 (±0.001)** | 0.011 (±0.002) | **0.005 (±0.000)** | **0.009 (±0.000)** | **0.062 (±0.001)** |
| SN | **0.004 (±0.000)** | 0.055 (±0.001) | 0.052 (±0.000) | 0.096 (±0.000) | **0.001 (±0.001)** | 0.050 (±0.004) | 0.044 (±0.001) | 0.091 (±0.004) |

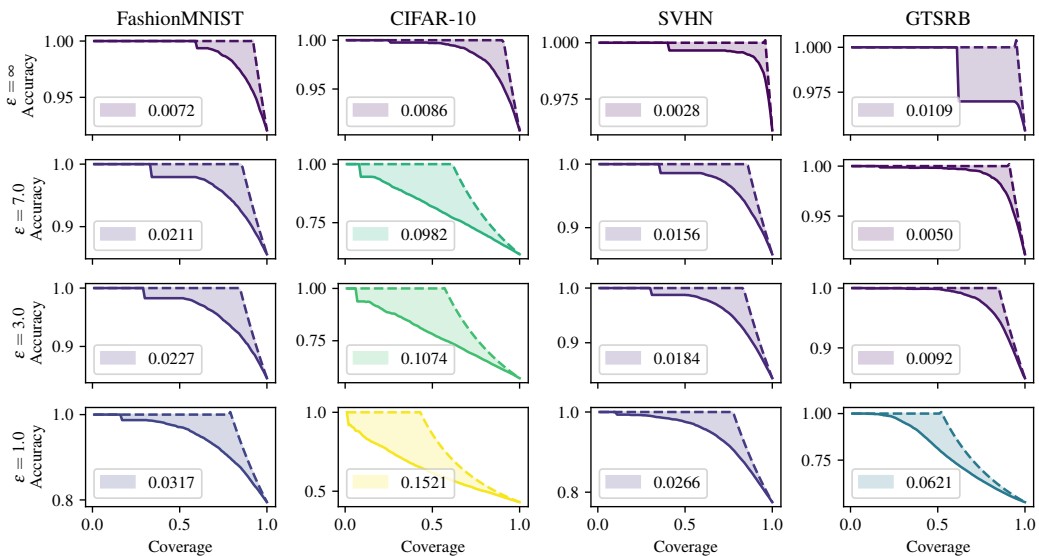

Figure 4: **Distance to accuracy-dependent upper bound for the** SCTD **method.** We plot both the accuracy-coverage trade-off of SCTD, i.e., $\mathrm{acc}_c(f, g)$, with a solid line and the upper bound $\overline{\mathrm{acc}}(a_{\mathrm{full}}, c)$ with a dashed line. The shaded region enclosed between the two curves corresponds to the accuracy-normalized SC score $s_{a_{\mathrm{full}}}(f, g)$. We observe that the score grows with stronger DP levels (shown by brighter colors), i.e., selective classification becomes harder at low $\varepsilon$s.

**Recovering non-private utility by reducing coverage.** Recall that one key motivation for applying selective classification to a DP model is to recover non-private model utility at the expense of coverage. We investigate this coverage cost in detail by examining how many samples a DP model can produce decisions for while maintaining the utility level of the respective non-private model (i.e., a model trained on the same dataset without DP). The results are outlined in Table 1. For high (i.e., $\varepsilon = 7$) and moderate (i.e., $\varepsilon = 3$) privacy budgets, we observe that a reduction of $20\% - 30\%$ in data coverage recovers non-private utility. However, at low (i.e., $\varepsilon = 1$) privacy budget we observe that dataset coverage degrades strongly on most datasets. In particular, we find that for CIFAR-10 (across all $\varepsilon$ values) and GTSRB (at $\varepsilon = 1$), the coverage reduces to below $30\%$. This result showcases that, depending on the particular choice of $\varepsilon$, recovering non-private model performance from a DP model can come at a considerable coverage cost. Moreover, this coverage cost varies widely across SC methods with SCTD leading to the lowest incurred loss in coverage by far.

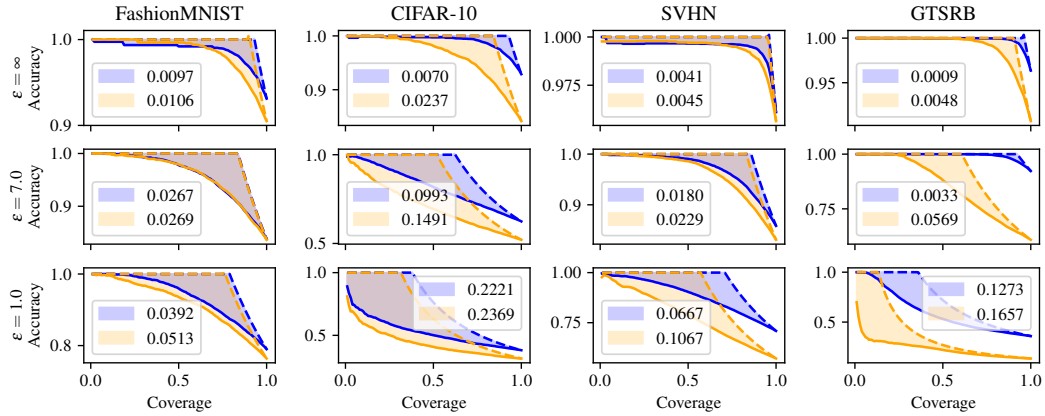

Figure 5: **Sequential composition (`DE`) vs parallel composition (`DE-PART`) for deep ensembles.**
We observe that `DE-PART` (orange) under-performs `DE` (blue) for selective classification across all $\varepsilon$.

**Disentangling selective prediction performance from base accuracy.** Recall from Section 3.4 that the standard evaluation pipeline for selective classification is not suitable for comparing DP models across varying privacy guarantees. To overcome this issue, we compute the accuracy-dependent upper bound (Equation 5) for each experiment and measure how closely the achieved accuracy/coverage trade-off aligns with this bound as per Equation 6. We document these results computed over all datasets, SC methods, and $\varepsilon$ levels in Table 2. We find that, as $\varepsilon$ decreases, all methods perform progressively worse. That is, for stronger privacy, all methods increasingly struggle to identify points they predict correctly on. Recall that this behavior is expected based on our discussion in Section 3.3. Again, we observe `SCTD` offering the strongest results, leading to the smallest bound deviation. To graphically showcase closeness to the upper bound, we further plot the accuracy-coverage trade-off and the corresponding upper bound for each experiment with the `SCTD` method in Figure 4.

**Parallel composition with partitioned ensembles.** As previously stated, Deep Ensembles under-perform under DP due to composition. One potential mitigation strategy is to partition the data and train isolated models with no data overlap. This circumvents composition but also limits the utility of individual ensemble members as less data is available for training. We experiment with both setups and report results in Figure 5. Overall, these experimental results indicate that (i) partitioned deep ensembles lead to lower full-coverage accuracy when compared to non-partitioned deep ensembles while (ii) at the same time being less performant in terms of selective classification performance.

## 5 Conclusion

In this work we have studied methods for performing selective classification under a differential privacy constraint. To that end, we have highlighted the fundamental difficulty of performing selective prediction under differential privacy, both via a synthetic example and empirical studies. To enable this analysis, we introduced a novel score that disentangles selective classification performance from baseline utility. While we establish that SC under DP is indeed challenging, our study finds that a specific method (`SCTD`) achieves the best trade-offs between SC performance and privacy budget.

**Limitations.** This work presents insights drawn from empirical results and we acknowledge that a more thorough theoretical analysis is needed for a deeper understanding of the interplay between SC and DP. Also, we did not carefully investigate fairness implications beyond class imbalance. This connection to fairness requires special attention with past works showing that both DP and SC can negatively affect sensitive subgroups [Jones et al., 2020, Bagdasaryan et al., 2019].

**Future work.** We believe that this work initiates a fruitful line of research. In particular, future work can expand on the intuition provided here to obtain fundamental bounds on selective classification performance in a DP setting. Although we did not focus on this here, we believe that the ideas developed in this paper could help mitigate the trade-offs between privacy and subgroup fairness.

## Acknowledgements

We would like to acknowledge our sponsors, who support our research with financial and in-kind contributions: CIFAR through the Canada CIFAR AI Chair, DARPA through the GARD project, NSERC through the COHESA Strategic Alliance and a Discovery Grant, and the Sloan Foundation. Anvith Thudi is supported by a Vanier Fellowship from the Natural Sciences and Engineering Research Council of Canada. Resources used in preparing this research were provided, in part, by the Province of Ontario, the Government of Canada through CIFAR, and companies sponsoring the Vector Institute. We would further like to thank Relu Patrascu at the University of Toronto for the provided compute infrastructure needed to perform the experimentation outlined in this work. Stephan Rabanser additionally wants to thank Mohammad Yaghini, David Glukhov, Patty Liu, and Mahdi Haghifam for fruitful discussions.

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

# A  Additional Method Details

## A.1  DP-SGD Algorithm

We provide a detailed definition of DP-SGD in Algorithm 2.

---

**Algorithm 2:** DP-SGD [Abadi et al., 2016]

---

**Require:** Training dataset $D$, loss function $\ell$, learning rate $\eta$, noise multiplier $\sigma$, sampling rate $q$, clipping norm $c$, iterations $T$.
1: **Initialize** $\theta_0$
2: **for** $t \in [T]$ **do**
3:     **1. Per-Sample Gradient Computation**
4:     Sample $B_t$ with per-point prob. $q$ from $D$
5:     **for** $i \in B_t$ **do**
6:         $g_t(\boldsymbol{x}_i) \leftarrow \nabla_{\theta_t} \ell(\theta_t, \boldsymbol{x}_i)$
7:     **end for**
8:     **2. Gradient Clipping**
9:     $\bar{g}_t(\boldsymbol{x}_i) \leftarrow g_t(\boldsymbol{x}_i)/\max\left(1, \frac{\|g_t(\boldsymbol{x}_i)\|_2}{c}\right)$
10:    **3. Noise Addition**
11:    $\tilde{g}_t \leftarrow \frac{1}{|B_t|}\left(\sum_i \bar{g}_t(\boldsymbol{x}_i) + \mathcal{N}(0, (\sigma c)^2 \mathbf{I})\right)$
12:    $\theta_{t+1} \leftarrow \theta_t - \eta \tilde{g}_t$
13: **end for**
14: **Output** $\theta_T$, privacy cost $(\varepsilon, \delta)$ computed via a privacy accounting procedure

---

## A.2  Selective Classification Method Details

**Softmax Response (SR)**  The traditional baseline methods for selective prediction is the *Softmax Response* (SR) method [Hendrycks and Gimpel, 2016, Geifman and El-Yaniv, 2017]. This method uses the confidence of the final prediction model $f$ as the selection score:

$$g_{\text{SR}}(\boldsymbol{x}, f) = \max_{c \in C} f(\boldsymbol{x}) \tag{7}$$

While this method is easy to implement and does not incur any additional cost, SR has been found to be overconfident on ambiguous, hard-to-classify, or unrecognizable inputs.

**SelectiveNet (SN)**  A variety of SC methods have been proposed that leverage explicit architecture and loss function adaptations. For example, *SelectiveNet* (SN) [Geifman and El-Yaniv, 2019] modifies the model architecture to jointly optimize $(f, g)$ while targeting the model at a desired coverage level $c_{\text{target}}$. The augmented model consists of a representation function $r : \mathcal{X} \rightarrow \mathbb{R}^L$ mapping inputs to latent codes and three additional functions: (i) the *prediction* function $f : \mathbb{R}^L \rightarrow \mathbb{R}^C$ for the classification task targeted at $c_{\text{target}}$; (ii) the *selection* function $g : \mathbb{R}^L \rightarrow [0, 1]$ representing a continuous approximation of the accept/reject decision for $\boldsymbol{x}$; and (iii) an additional *auxiliary* function $h : \mathbb{R}^L \rightarrow \mathbb{R}^C$ trained for the unconstrained classification tasks. This yields the following losses:

$$\mathcal{L} = \alpha \mathcal{L}_{f,g} + (1 - \alpha)\mathcal{L}_h \tag{8}$$

$$\mathcal{L}_{f,g} = \frac{\frac{1}{M}\sum_{m=1}^{M} \ell(f \circ r(\boldsymbol{x}_m), y_m)}{\text{cov}(f, g)} + \lambda \max(0, c - \text{cov}(f, g))^2 \tag{9}$$

$$\mathcal{L}_h = \frac{1}{M}\sum_{m=1}^{M} \ell(h \circ r(\boldsymbol{x}_m), y_m) \tag{10}$$

The selection score for a particular point $\boldsymbol{x}$ is then given by:

$$g_{\text{SN}}(\boldsymbol{x}, f) = \sigma(g \circ r(\boldsymbol{x})) = \frac{1}{1 + \exp(g \circ r(\boldsymbol{x}))} \tag{11}$$

**Self-Adaptive Training (SAT)**  Alternatively, prior works like Deep Gamblers [Liu et al., 2019] and *Self-Adaptive Training* [Huang et al., 2020] have also considered explicitly modeling the abstention

class $\perp$ and adapting the optimization process to provide a learning signal for this class. For instance, *Self-Adaptive Training* (SAT) incorporates information obtained during the training process into the optimization itself by computing and monitoring an exponential moving average of training point predictions over the training process. Samples with high prediction uncertainty are then used for training the abstention class. To ensure that the exponential moving average captures the true prediction uncertainty, an initial burn-in phase is added to the training procedure. This delay allows the model to first optimize the non-augmented, i.e., original $C$-class prediction task and optimize for selective classification during the remainder of the training process. The updated loss is defined as:

$$\mathcal{L} = -\frac{1}{M} \sum_{m=1}^{M} \left( t_{i,y_i} \log p_{i,y_i} + (1 - t_{i,y_i}) \log p_{i,C+1} \right) \tag{12}$$

The rejection/abstention decision is then determined by the degree of confidence in the rejection class:

$$g_{\texttt{SAT}}(\boldsymbol{x}, f) = f(\boldsymbol{x})_{C+1} \tag{13}$$

**Deep Ensembles (DE)**    Finally, ensemble methods combine the information content of $M$ models into a single final model. Since these models approximate the variance of the underlying prediction problem, they are often used for the purpose of uncertainty quantification and, by extension, selective prediction. The canonical instance of this approach for deep learning based models, *Deep Ensembles* (DE) [Lakshminarayanan et al., 2017], trains multiple models from scratch with varying initializations using a proper scoring rule and adversarial training. Then, after averaging the predictions made by the model, the softmax response (SR) mechanism is applied:

$$g_{\texttt{DE}}(\boldsymbol{x}, f) = \max_{c \in C} \frac{1}{M} \sum_{m=1}^{M} f_{\boldsymbol{\theta}_{m,T}}(\boldsymbol{x}). \tag{14}$$

**Monte-Carlo Dropout (MCDO)**    To overcome the computational cost of estimating multiple models from scratch, *Monte-Carlo Dropout* (MCDO) [Gal and Ghahramani, 2016] allows for bootstrapping of model uncertainty of a dropout-equipped model at test time. While dropout is predominantly used during training to enable regularization of deep neural nets, it can also be used at test time to yield a random sub-network of the full neural network. Concretely, given a model $f$ with dropout-probability $o$, we can generate $M$ random sub-networks at test-time by deactivating a fraction $o$ of nodes. For a given test input $\boldsymbol{x}$ we can then average the outputs over all models and apply softmax response (SR):

$$g_{\texttt{MCDO}}(\boldsymbol{x}, f) = \max_{c \in C} \frac{1}{Q} \sum_{q=1}^{Q} f_{o(\boldsymbol{\theta}_T)}(\boldsymbol{x}) \tag{15}$$

**Selective Classification Training Dynamics (SCTD)**    Another ensembling approach that has recently demonstrated state-of-the-art selective classification performance is based on monitoring the model evolution during the training process. *Selective Classification Training Dynamics* (SCTD) [Rabanser et al., 2022] records intermediate models produced during training and computes a disagreement score of the intermediate predictions with the final prediction for any test-time input $\boldsymbol{x}$:

$$g_{\texttt{SCTD}}(\boldsymbol{x}, f) = \sum_{t=1}^{T} v_t a_t(\boldsymbol{x}, f) \quad \text{with} \quad a_t(\boldsymbol{x}, f) = \begin{cases} 1 & f_{\boldsymbol{\theta}_t}(\boldsymbol{x}) \neq f_{\boldsymbol{\theta}_T}(\boldsymbol{x}) \\ 0 & \text{otherwise} \end{cases} \quad v_t = \left( \frac{t}{T} \right)^k \tag{16}$$

SCTD only observes the training process and does not modify the model architecture/training objective.

## B    Additional Experimental Details

### B.1    Hyperparameters

In this section, we document additional hyper-parameter choices. For Self-Adaptive Training (SAT), we set the pre-training epochs to $100$ and momentum parameter $0.9$. For Selective Classification Training Dynamics (SCTD), we set the weighting parameter $k = 3$ and consider checkpoints at a $50$ batch resolution. For Monte-Carlo Dropout, we set the dropout probability to $0.1$. Entropy regularization as suggested in Feng et al. [2023] is employed with $\beta = 0.01$.

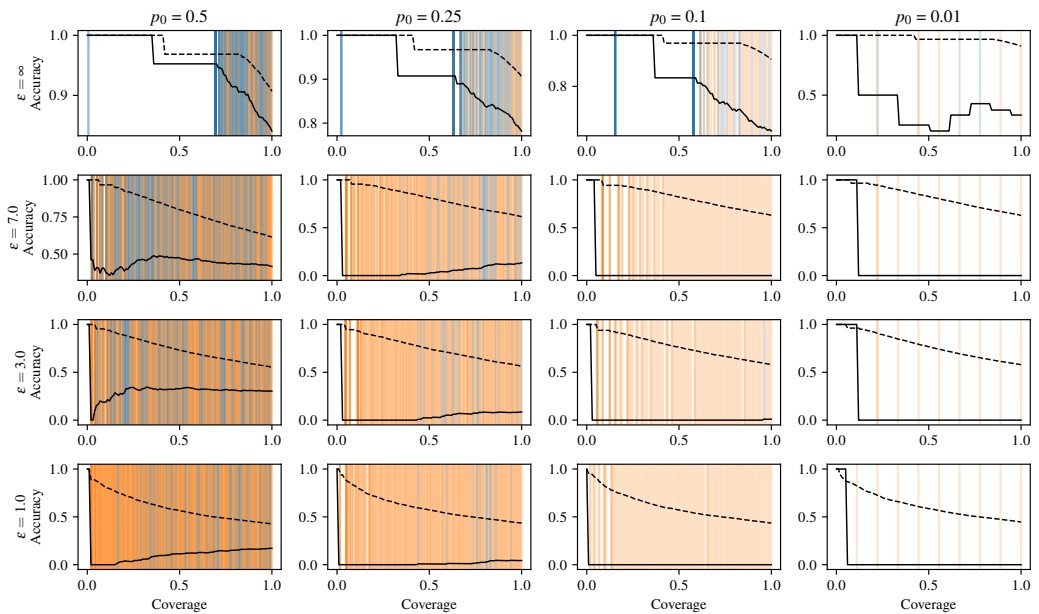

Figure 6: **Inducing a class imbalance on CIFAR-10**. We train multiple CIFAR-10 models across privacy levels and sampling probabilities for class 0 given by $p_0$. We plot the accuracy/coverage trade-off as well at the exact coverage level at which any point from the minority class is accepted. The accuracy-coverage trade-off for the full dataset is given by the dashed line while the trade-off for the minority group only is given by the solid line. Blue vertical lines show correctly classified points, orange points show incorrectly classified points. Non-private models accept correct points first and do so at the end of the coverage spectrum (i.e., majority points are accepted first). As we increase privacy (i.e., decrease $\varepsilon$), the model is increasingly unable to rank minority examples based on prediction correctness and even accepts incorrect points first. Moreover, the accuracy of the model on the minority class decreases with stronger DP. These effects are especially strong for small sampling probabilities.

## B.2 Class Imbalance Experiments

We provide additional experiments on the effect of class imbalance to extend our intuition from Section 3.3. To that end, we take two data sets from our main results, namely CIFAR-10 and FashionMNIST, and produce four alternate datasets from each dataset. These datasets feature various degrees of class imbalance with $p_0 \in \{0.5, 0.25, 0.1, 0.01\}$ specifying the sampling probability for class 0. All other classes maintain a sampling probability of 1. We then train the same model as described in Section 4 and apply the softmax response SC algorithm.

We document these results in Figures 6 and 7. For $\varepsilon = \infty$, we observe the expected gains from SC: minority points are accepted towards the end of the coverage spectrum [Jones et al., 2020] and correct points are accepted first. This effect is independent of the sampling probability $p_0$. As we decrease the $\varepsilon$ budget we observe that (i) the acceptance of minority groups starts to spread over the full coverage spectrum; (ii) the accuracy on the subgroup increasingly deteriorates with smaller $\varepsilon$, and (iii) wrongful overconfidence on the minority reverses the acceptance order at low sampling probabilities $p_0$ (i.e., incorrect points are often accepted first). These results indicate that employing selective classification on private data can have unwanted negative effects in the presence of subgroups. Future work should investigate this connection more thoroughly.

## B.3 Upper Bound Reachability

In Equation 5, we have introduced an upper bound on the selective classification performance on a model with full-coverage accuracy $a_{\text{full}}$. We now present a simple experimental panel across varying

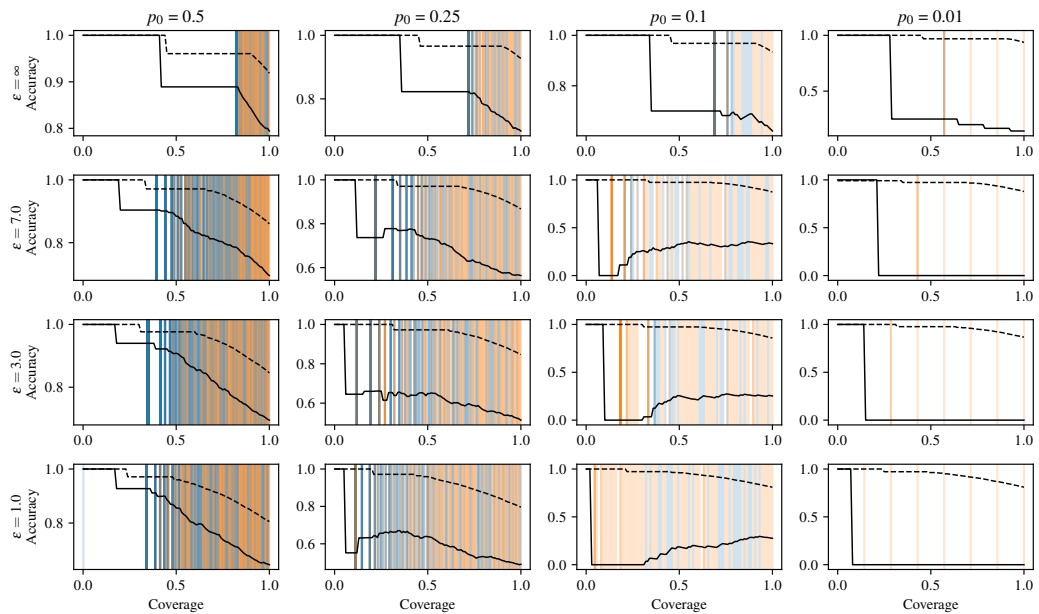

Figure 7: **Inducing a class imbalance on FashionMNIST**. Same insights as in Figure 7.

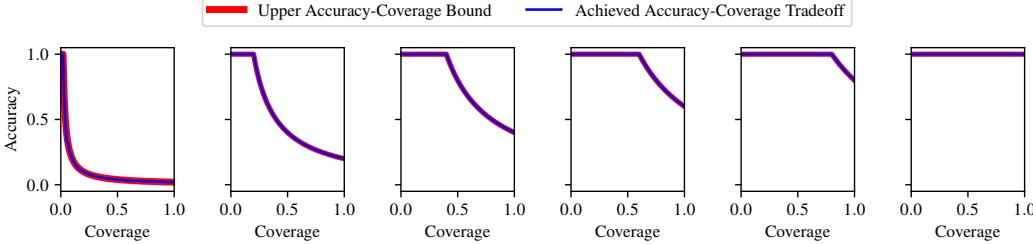

Figure 8: **Upper bound matching experiment**. The experiment as described in Section B.3 matches the optimal bound exactly across multiple full-coverage accuracy levels.

full-coverage accuracy levels showing that this bound is in fact reachable by a perfect selective classifier.

We assume a binary classification setting for which we generate a true label vector $\boldsymbol{y} \in \{0,1\}^{n_0+n_1}$ with balanced classes, i.e., $n_0 = n_1$ where $n_0$ corresponds to the number points labeled as $0$ and $n_1$ corresponds to the number points labeled as $1$. Then, based on a desired accuracy level $a_{\text{full}}$, we generate a prediction vector $\boldsymbol{p}$ which overlaps with $\boldsymbol{y}$ for a fraction of $a_{\text{full}}$. Finally, we sample a scoring vector $\boldsymbol{s}$ where each correct prediction is assigned a score $s_i \sim \mathcal{U}_{0,0.5}$ and each incorrect prediction is assigned a score $s_i \sim \mathcal{U}_{0.5,1}$. Here, $\mathcal{U}_{a,b}$ corresponds to the uniform distribution on the interval $[a, b)$. This score is clearly optimal since thresholding the scoring vector $\boldsymbol{s}$ at $0.5$ perfectly captures correct/incorrect predictions: all $s_i < 0.5$ correspond to a correct prediction, while all $s_i \geq 0.5$ correspond to an incorrect prediction. Computing the accuracy/coverage trade-off of this experiment, across various utility levels, matches the bound exactly.

