# OpenReview forum: "Training Private Models That Know What They Don’t Know"
_NeurIPS.cc/2023/Conference — NeurIPS 2023 poster_

### Official Review · Reviewer_rhFq · 2023-06-27

**Soundness:** 2 fair
**Presentation:** 3 good
**Contribution:** 2 fair
**Rating:** 5
**Confidence:** 3

**Summary:**

This paper explores the combination of differential privacy (DP) and selective classification. It examines the performance variations of existing selective classification methods when equipped with DP-SGD. Additionally, the paper demonstrates the complex correlation between selective accuracy and differential privacy. To address the challenge of disentangling selective classification performance from baseline utility, a new metric is proposed to evaluate different selective classification algorithms under differential privacy. The paper concludes by highlighting the superior performance of the recent algorithm SCTD in DP settings.

**Strengths:**

The paper explores an intriguing intersection between differential privacy and selective classification. The experiments conducted are comprehensive.

**Weaknesses:**

One potential weakness of this work lies in the exclusive use of DP-SGD for all the differential privacy implementations, which could be a subject of questioning. There exist various methods to enforce differential privacy, such as perturbing output scores ($g(x,f)$) or perturbing training data ($x,y$). In the case of ensembling, applying the DP mechanism in the softmax step, as shown in [1], could be considered. Consequently, ensembling methods do not necessarily suffer from DP composition. Therefore, it might be an overstatement to claim that SCTD performs the best in a private learning setting; instead, it can be stated that it performs well with a straightforward adaptation of DP-SGD.

[1] Phan, NhatHai, et al. "Differential privacy preservation for deep auto-encoders: an application of human behavior prediction." Proceedings of the AAAI Conference on Artificial Intelligence. Vol. 30. No. 1. 2016.

**Questions:**

- The term "full-coverage accuracy" is repeatedly referenced without a formal definition. It is unclear whether it represents an arbitrary number within the range of [0,1].
- The advantage of Definition 3.2 is not adequately explained. Since integration is additive, Equation (6) seems to yield a negative AUC score plus a constant determined only by $a_{full}$. This may not provide more informative insights than the original AUC score. Consequently, the benefits of this new metric remain unclear. Moreover, Definition 3.2 is rather confusing. Is $a_{full}$ a variable or a fixed value determined by $f,g$? If it is a variable, how is it chosen among different algorithms? If it is algorithm-dependent, then this is just the (negative) original AUC score penalized by a function of the full-coverage accuracy. It would be helpful to elaborate further on the motivation behind this metric.

**Limitations:**

Yes.

---

> ### Author Rebuttal · Authors · 2023-08-06
>
> We thank the reviewer for their feedback on our work and address individual concerns below.
>
> > One potential weakness of this work lies in the exclusive use of DP-SGD for all the differential privacy implementations, which could be a subject of questioning. There exist various methods to enforce differential privacy, such as perturbing output scores $(g(x,f))$ or perturbing training data $(x,y)$.
>
> While we agree with the observation that differential privacy can be ensured in a variety of ways, DP-SGD has become the de-facto standard for training differentially private deep learning models. This is in part due to its ability to be applied to non-convex models with arbitrary architectural modifications and loss functions. In contrast, output perturbation is only useful for convex models, and we are dealing with non-convex models inherently in our work. Hence, the only algorithms at our disposal are noisy first order or second order methods. We further emphasize that selective prediction is especially important for deep learning models as the robustness of deep neural nets constitutes a major open problem. Since DP-SGD is the most flexible and widely used approach to ensure privacy in deep learning, we believe that it is most important to focus our work on the interplay between selective classification and DP-SGD in particular.
>
> > In the case of ensembling, applying the DP mechanism in the softmax step, as shown in [1], could be considered. Consequently, ensembling methods do not necessarily suffer from DP composition. Therefore, it might be an overstatement to claim that SCTD performs the best in a private learning setting; instead, it can be stated that it performs well with a straightforward adaptation of DP-SGD.
>
> The work presented in [1] considers the functional mechanism to ensure $\epsilon$-DP. However, the noise injection from the functional mechanism (noising weights and objective perturbation) requires strong convexity assumptions [2]. This precludes the functional mechanism to be applied directly to deep neural networks which feature non-convex loss landscapes. As we argue in the previous point, the focus of our work is to study the interplay of differential privacy and selective classification for modern deep neural nets.
>
> [2]: Ponomareva, Natalia, et al. "How to dp-fy ml: A practical guide to machine learning with differential privacy." Journal of Artificial Intelligence Research 77 (2023): 1113-1201.
>
> > The term "full-coverage accuracy" is repeatedly referenced without a formal definition. It is unclear whether it represents an arbitrary number within the range of [0,1].
>
> Full-coverage accuracy refers to the accuracy of a trained model on the entire (i.e., full) test set, formally $a_\text{full} = \text{acc}_{c=1}(f,g)$ (using the definition from Equation 4). We have made sure to state this explicitly in our revision to avoid ambiguity.
>
> > The advantage of Definition 3.2 is not adequately explained. Since integration is additive, Equation (6) seems to yield a negative AUC score plus a constant determined only by $a_\text{full}$. This may not provide more informative insights than the original AUC score. Consequently, the benefits of this new metric remain unclear. Moreover, Definition 3.2 is rather confusing. Is $a_\text{full}$ a variable or a fixed value determined by $(f,g)$? If it is a variable, how is it chosen among different algorithms? If it is algorithm-dependent, then this is just the (negative) original AUC score penalized by a function of the full-coverage accuracy. It would be helpful to elaborate further on the motivation behind this metric.
>
> The advantage of the accuracy-normalized selective classification score $s_{a_\text{full}}(f,g)$ from Definition 3.2 is the fact that it constitutes a performance measure free from accuracy alignment. As we explain in the second paragraph of Section 3.4, accuracy alignment is not possible under DP which necessitates a new performance metric. This new metric, $s_{a_\text{full}}(f,g)$ computes how close the observed accuracy-coverage tradeoff aligns with the computed upper bound on the selective classification performance (Equation 5). As such it captures the area enclosed between the achieved accuracy-coverage tradeoff and the corresponding upper bound. As we state in Definition 3.1, $a_\text{full}$ is fixed (also see response to previous point) and determined by $(f,g)$. This performance metric allows us to compare selective classification performance across privacy levels and further quantifies closeness to a perfect selective classifier.
>
> It would be appreciated if the reviewer could elaborate on why they believe that the score computed in Equation 6 can be negative. Since we subtract the actual achieved accuracy at a certain coverage level from the computed upper bound at that coverage level, it is ensured that the integrand is non-negative. Subsequent integration of non-negative integrands further guarantees non-negativity.
>
> ---
> **We hope that we have addressed the reviewer’s concerns and that the reviewer considers raising their score as a result.**

---

> > ### Comment · Reviewer_rhFq · 2023-08-15
> > **Thank you for the rebuttal**
> >
> > Thanks for the reply. The rebuttal have addressed some of my concerns. However, I am still not totally convinced by the justification of Eq. (6). To clarify, I did not say the score can be negative. What I meant is that the integration in Eq. (6) can be decomposed into two parts $A-B$, where term $A$ is merely a function of $a_{full}$ and term $B$ is just the AUC. In other words, for two algorithms with the same $a_{full}$, comparing the metric in Eq. (6) is just equivalent to comparing the negative AUC.
> >
> > If $a_{full}$ is determined by the selective classification algorithm, then I may get a bit of this definition, which makes sense for normalization.  I have updated my evaluation accordingly.

---

> > > ### Author Response · Authors · 2023-08-15
> > > **Additional clarification**
> > >
> > > We thank the reviewer for reconsidering their evaluation given our rebuttal and for sharing their additional concerns with us. We address these concerns below.
> > >
> > > > However, I am still not totally convinced by the justification of Eq. (6). To clarify, I did not say the score can be negative. What I meant is that the integration in Eq. (6) can be decomposed into two parts $A-B$, where term $A$ is merely a function of $a_\text{full}$ and term $B$ is just the AUC. In other words, for two algorithms with the same $a_\text{full}$, comparing the metric in Eq. (6) is just equivalent to comparing the negative AUC.
> > >
> > > We apologize for misunderstanding the reviewer’s claim about the negativity of our score. The reviewer is right that in the case of two algorithms with the same $a_\text{full}$ it is sufficient to compare their AUC scores. However, this equivalence is intentional and shows that our performance metric in Equation 6 reduces to the already established performance metric (Equation 4) under accuracy alignment. In the less trivial case of misaligned full-coverage accuracies (where now $a_\text{full}$ does not cancel), which is unavoidable when evaluating DP models across $\varepsilon$-levels, Equation 6 allows us to disentangle utility gains from gains in selective classification performance. Note that disentanglement is not possible when comparing AUC scores computed via Equation 4 across varying full-coverage accuracies as the score is biased towards higher utility models. In short, Equation 6 can handle both the accuracy-aligned setting as well as the non-accuracy-aligned setting and therefore offers strictly more flexibility in evaluating selective classification.
> > >
> > > We further remind the reviewer that Equation 6 computes the discrepancy between the actual achieved accuracy-coverage tradeoff and an upper bound on the accuracy-coverage tradeoff assuming a perfect acceptance ordering under the $a_\text{full}$ constraint. That is, it computes a measure of selective classification performance regret. This enables us to not just compare performance across models but also to quantify the distance to an idealized selective classifier. If Equation 6 evaluates to 0, then the selective classifier has achieved perfect SC performance (under the $a_\text{full}$ constraint as determined by $(f,g)$). In contrast, the AUC computed in Equation 4 reaches its optimal value of 1 iff a classifier is 100% accurate (which alleviates the need for selective classification altogether). As such, the AUC score conflates gains in accuracy with gains in SC performance which is undesirable for methods that want to isolate SC performance.
> > >
> > > To strengthen this intuition, we would like to point the reviewer to Appendix B.3 and Figure 6 which shows SC bounds for a variety of full-coverage accuracy levels on a synthetic dataset. The rightmost panel shows the tradeoff obtained by a perfect classifier at 100% accuracy at 100% coverage. The AUC score (Equation 4) assigns the best score to this setting. Panels further left show the bound at increasingly lower full-coverage accuracy levels (which leads to a decrease in AUC). However, all of the depicted panels show the accuracy-coverage tradeoff of a perfect selective classifier under the respective full-coverage accuracy constraint. For example, the second panel shows the tradeoff for a model that is 20% accurate on the entire test set. Despite this constraint on the accuracy, the depicted trade-off is optimal as it allows for the acceptance of all 20% correct points first and accepting all 80% of incorrect points last. Our score from Equation 6 allows us to quantify closeness to this optimal setting. Conversely, as a result of low accuracy, a perfect classifier under the 20% accuracy constraint (i.e. a selective classifier matching our bound) only has an AUC of 0.52 which might erroneously suggest that there is additional room for improvement of SC performance. This shows that well performing selective classifiers cannot be characterized by relying on AUC (Equation 4) alone and that Equation 6 uncovers and mitigates this failure mode.
> > >
> > > We are happy to include this discussion as part of the revised appendix. Moreover, if the reviewer would benefit from a graphical depiction showcasing the example mentioned in the previous paragraph in more detail, then we are happy to share a plot with the AC (who can then share the plot with the reviewer at their own discretion).
> > >
> > > ---
> > > **We hope that we have addressed the reviewer’s additional concerns and are happy to further engage with them in case they still desire further clarification.**

---

### Official Review · Reviewer_atra · 2023-07-04

**Soundness:** 3 good
**Presentation:** 4 excellent
**Contribution:** 3 good
**Rating:** 7
**Confidence:** 4

**Summary:**

This paper investigates the relationship between selective classification (SC) and differential privacy (DP).
First, the authors consider various SC approaches and if they incur additional privacy/utility costs under the constraint of DP.
They then investigate the effect of DP on SC algorithms and find that DP can drastically reduce the efficacy of SC as models become overconfident in the wrong class under DP.
Finally, the authors introduce a new metric to evaluate SC under DP that takes into account the baseline accuracy of the model.
An extensive experimental evaluation is given, showing which SC techniques perform best, the coverage needed to provide non-private accuracy, and an evaluation under the new metric.
Overall, SCTD (an SC approach that abstains based on the model's disagreement at various checkpoints) is shown to perform the best.

**Strengths:**

- This is a very interesting problem and a clever way to give private learning a boost in accuracy. The coverage required for the non-private accuracy table, in particular, was quite interesting.
- The experimental evaluation contains many different SC techniques. This seems like a large implementation effort and a nice contribution to the community.
- The example with class imbalance gave a nice intuition for the problem with SC under DP.
- The SCTD approach is a nice way to make use of intermediate checkpoints. It is often argued these intermediate model checkpoints are a waste of privacy budget or assume too strong of an adversary. It was nice to see them actually utilized here to improve classification accuracy.

**Weaknesses:**

## Groupings of SC techniques
The discussion of why the SC techniques were grouped as post-processing or composition was non-existent. There was discussion of why the DE technique affects privacy/utility, but none of the other approaches received any discussion as to why they were ``free" or had an effect.

I believe the SN approach was grouped with composition as it would also affect utility by changing the loss function (but not privacy, as DP-SGD is applied to the gradients, not the loss). But, the authors do not make this argument. Furthermore, by my previous logic, I expect DG or SAT to affect utility as they also modify the loss function. In general, this section on grouping the techniques had the least justification. I would like to see more justification here. It would also be beneficial to explain why specific approaches are free, as I think post-processing is an oversimplification (as some require modifying the training algorithm).

## DE evaluation
While I agree the DE approach affects privacy (needing multiple models), I wasn't convinced that composition was the only way to fix this problem. I was curious what would happen if one instead portioned the dataset between the models in the ensemble and used the parallel composition theorem. I realize this would decrease utility, but how would that compare to the composition?

## Delta too high
$1/n$ is too large for a delta parameter. It technically allows a trivial mechanism that publishes a single record of the database. The general rule of thumb is that $\delta << 1/n$ [Dwork & Roth](https://www.nowpublishers.com/article/Details/TCS-042). I don't believe changing the delta would affect the overall conclusions of the paper, so I am willing to let it go, but I wanted to bring attention to it.

## Typos/Clarity
Overall this paper was incredibly well written. However, below I give a couple of trouble spots that could be made more clear.
- Line 64: "more stringent than targeted privacy level" is clunky/hard to parse
- Line 124: Overloading the notation of $acc_c$ was rather confusing. I am not sure if this is normal in the SC literature, but having $c$ represent both a threshold and coverage with no clear distinction is rather confusing.
- Line 203: Incorrect acronym for deep ensembles.
- Line 215: mechanisms? or models? A mechanism is used to train a model, so I think the model makes more sense.
- Line 229: This first sentence is somewhat confusing. Specifically, "free from its bias towards accuracy is to align..." is a lot to parse.
- Line 246: It should be stated somewhere that this metric lower is better (as opposed to the previous metric that higher is better). To save the reader having to figure that out (although it is easy to figure out).
- Figure 2: not sure if error bars could be added here, although it might make it too hard to read
- Table 1: it should be made clear what the $\pm$ represents. I assume a confidence interval but at what $\alpha$?
- In Tables 1 & 2, the labels seem quite off. First, there are six techniques, but only five appear in Table 1. Second, MSP and NNTD are not defined. I assume NNTD = SCTD and MSP=SR?


**Questions:**

- Can the authors comment on the DE approach using the dataset partitioning idea? Perhaps they have tried it.
- Although SC does not affect the privacy guarantee, does it increase privacy leakage? For example, could a membership inference attack use the absence of a prediction to rule out non-member samples and increase overall attack performance?


**Limitations:**

The authors do a good job of acknowledging limitations. I agree the initial experimental evaluation is a significant contribution, but theoretical evaluations are important for future work.

---

> ### Author Rebuttal · Authors · 2023-08-06
>
> We thank the reviewer for their detailed assessment of our work and address individual concerns below.
>
> > Grouping of SC techniques
>
> The main reason for a method to fall into the composition category is that these methods require multiple passes over the data (and hence lead to a higher needed privacy budget). While most methods SC methods modify the loss or the architecture, DP-SGD is an optimizer and protects privacy independent of the exact loss function. SelectiveNet is a notable exception because it requires a fixed targeted coverage level at training time. However, in order to obtain a full characterization of the accuracy-coverage spectrum (which is needed for comparison with other methods), we have to train multiple models. This is also the reason why Figure 2 shows the performance of SelectiveNet with discrete dots and dashed lines. We have made sure to clarify the reasons for the grouping in Section 3.2.
>
> > DE evaluation
>
> This is an interesting and relevant suggestion! The reviewer is right that partitioning the dataset and training isolated models on these subsets would bypass composition and alleviate the need for lowering $\varepsilon$ on a per-ensemble-member basis. However, as also rightfully pointed out by the reviewer, the partitioning would heavily decrease overall model utility. To showcase this effect, we ran our full experimental panel again but partitioned the dataset into 5 subsets and trained the same set of DP models as in the main experimental section of the paper. We provide a corresponding figure in the PDF document associated with this rebuttal (Figure 1). This figure follows the same style as Figure 3 in the main paper but shows the accuracy-coverage tradeoff and upper-bound closeness for both standard deep ensembles ($\texttt{DE}$) and partitioned deep ensembles $\texttt{DE-PART}$.
>
> Overall, these experimental results indicate that (i) the partitioned deep ensembles (orange) lead to lower full-coverage accuracy when compared to non-partitioned deep ensembles (blue) while (ii) at the same time being less performant in terms of selective classification performance. We have included this result together with a discussion in the appendix of the paper.
>
> > Delta too high
>
> Thanks for catching this! Our experimental assumptions were based on [2], which claims that for a dataset with order $10^N$ samples $\delta$ should be set to $\delta = 10^{-N}$. We concur that this value should have been set to $\delta = 10^{-(N+1)}$ instead, as seems to be more common in the literature. We have run our experimental panel again with this setting and are happy to report that our results continue to hold even with a smaller delta.
>
> > Line 64: [...]
>
> We have rephrased this sentence as follows: In particular, accuracy-aligning multiple DP models via early-stopping frequently leads to a privacy level that is more stringent than targeted by the DP mechanism, rendering the intended comparison impossible.
>
> > Line 124: [...]
>
> We clarify that $c$ always refers to a specified (desired) coverage level while $\text{cov}_\tau(f,g)$ refers to the computed coverage at $\tau$.
>
> > Line 229: [...]
>
> We have rephrased this sentence as follows: The default approach of using the SC performance metric introduced in Equation 4 without its accuracy bias is to align different SC approaches/models at the same accuracy.
>
> > Line 246: [...]
>
> We have added a corresponding remark to the caption of Table 2 to make this more obvious to the reader.
>
> > Figure 2: [...]
>
> We experimented with adding an additional shaded region to emphasize seed randomness but indeed found that it would drastically decrease reliability of the plot. We do however provide $1 \cdot \sigma$ regions around all mean results across all experimental tables.
>
> > Table 1: [...]
>
> As we state in lines 295-296, we conduct all experiments over a total of 5 random runs. Tables 1 and 2 therefore report mean results over all random runs with a $1 \cdot \sigma$ interval (with sigma being the standard deviation over the random runs). We have made sure to emphasize this in the captions of Tables 1 and 2.
>
> > Line 203: [...]
>
> > Line 215: [...]
>
> > In Tables 1 & 2, the labels seem quite off. [...]
>
> We fixed these in our revised paper.
>
> > Although SC does not affect the privacy guarantee, does it increase privacy leakage? [...]
>
> This is a very interesting question! In fact, a similar question has been preliminarily explored in the Private Aggregation of Teacher Ensembles (PATE) [1] class of models. As a short summary, PATE trains an ensemble of teacher models and transfers the knowledge to a student model. Privacy is provided by training teachers on disjoint data and by a noisy aggregation of teacher answers. A rejection mechanism is employed on the class voting histogram [2, Alg 1]. A point is rejected when there is large disagreement in the noised aggregation of predictions across teachers, otherwise the noised prediction is released. The privacy budget is spread over both the initial consensus checking phase and the prediction release phase. Hence, some points only incur the rejection privacy budget while other points incur both the rejection check and prediction release budget. A rejection decision therefore can therefore provide some signal that could be used by an MI adversary.
>
> That said, it is not evidently clear how to apply the same chain of reasoning to models that are trained using DP-SGD (which has a single $\epsilon$ budget). Although membership inference and selective classification often rely on very similar mechanisms (e.g. logit or softmax thresholding), the precise connection between MI (and privacy leakage in general) and selective classification has not been established yet. This should be investigated in future work.
>
> **References:**
>
> [1]: Papernot, Nicolas, et al. "Semi-supervised knowledge transfer for deep learning from private training data." ICLR 2017.
>
> [2]: Papernot, Nicolas, et al. "Scalable private learning with pate." ICLR 2018.

---

> > ### Comment · Reviewer_atra · 2023-08-15
> > **Reply to Rebuttal**
> >
> > I want to thank the Authors for a very detailed response to the reviews.
> >
> > - Thanks for clarifying the grouping; this makes much more sense. I would still argue post-processing is not quite the correct name, as changing the utility function/architecture can not be done after training, but I am ok with it (now it is more clearly explained).
> > - I really appreciated the additional partitioned DE evaluation and am convinced by the results.
> > - Also, thanks for the discussion on MI. I agree it will be interesting to pursue this connection further (but out of the scope of this work).
> >
> > Overall I am satisfied with the response and have updated my score.

---

> > > ### Author Response · Authors · 2023-08-16
> > > **Thank you**
> > >
> > > We thank the reviewer for considering our rebuttal and for raising their score! We were happy to hear that their concerns have been addressed.
> > >
> > > We provide one last point of clarification wrt the post-processing terminology. DP-SGD ensures privacy by clipping and noising individual gradients. The output of this private mechanism is a set of gradient updates that are subsequently used for model training. Any selective classification mechanism that computes a function derived from a trained DP-SGD model (or its intermediate models) therefore falls under the post-processing property of DP: If $\phi(x)$ satisfies $(\varepsilon,\delta)$-DP, then for any deterministic or randomized function $\psi(\cdot)$,$\psi \circ \phi (x)$ satisfies $(\varepsilon,\delta)$-DP. In our case $\phi(\cdot)$ corresponds to the training process of a single model via DP-SGD and $\psi(\cdot)$ corresponds to the application of a selection/rejection mechanism. We hope this makes our justification for using the term post-processing clearer.

---

> > > > ### Comment · Reviewer_atra · 2023-08-17
> > > > **Reply**
> > > >
> > > > Thanks for the final clarification. I personally always viewed the output of DP-SGD as a trained model, not gradients. I see your reasoning now, Thanks.

---

### Official Review · Reviewer_sqFp · 2023-07-05

**Soundness:** 3 good
**Presentation:** 1 poor
**Contribution:** 2 fair
**Rating:** 6
**Confidence:** 2

**Summary:**

This paper explores the interaction between differential privacy and selective classification. Several methods for training selective classifiers are considered and evaluated for their compatibility with DP-SGD. Experimental results are given that show decreasing privacy parameters leads to selective classifiers becoming increasingly confident in incorrect predictions. An alternative performance metric for selective classification is proposed for the purposes of evaluating DP selective classifiers.

**Strengths:**

The paper, to the best of my knowledge, is the first to empirically investigate the degradation of performance of selective classifiers under the constraint of differential privacy. It addresses interesting questions regarding the training and evaluation of DP selective classifiers, and highlights important distinctions between SC training methods that become particularly relevant in the setting of differential privacy.

**Weaknesses:**

I think additional attention to presentation is necessary to clarify the main contributions of the paper. The punchline of the experiment in the body of the paper is that the observed overconfidence in an incorrect classification prevents selective classification, but this is an underspecified claim. I take it to mean that highly accurate selective classification based on prediction confidence is in tension with strong privacy guarantees, but this is consistent with known privacy/accuracy tradeoffs and doesn't seem terribly surprising. The necessity for new evaluation metrics is also still unclear to me. The metric given in equation 4 is an area under the curve of accuracy as a function of coverage, as coverage ranges from 0 to 1, but then the stated motivation for the new evaluation metric is that we want to avoid aligning selective classifiers based on accuracy. As stated, it doesn't seem like equation 4 is aligning on accuracy, and that training two selective classifiers with different privacy parameters would allow for a meaningful comparison by (an approximation) of equation 4.



**Questions:**

My main questions are reiterations of earlier critique. Why do existing evaluation metrics fail in the case of private selective classification? Is there additional significance to the empirical results beyond the extension of privacy/utility tradeoffs to the case of selective classification?

In Figure 1, it would be nice to have the meaning of the various colors used explicitly stated, ideally in the caption. It's inferable from context, but additional description would probably save the reader time. I also found the results in figure 4 and 5 were challenging to parse, mostly the significance of the blue and orange lines. Additional explanatory text to help the reader interpret these lines would be greatly appreciated.

**Limitations:**

Yes, the authors have sufficiently addressed the impact of their work.

---

> ### Author Rebuttal · Authors · 2023-08-06
>
> We thank the reviewer for their feedback on our work and address individual concerns below.
>
> > I think additional attention to presentation is necessary to clarify the main contributions of the paper. [...]
>
> To the best of our knowledge, and as the reviewer acknowledges themselves in the _Strengths_ section, our work is the first to examine the connection between privacy and selective prediction. In particular, we examine both (i) how DP constraints can affect SC performance; and (ii) how SC methods in turn affect DP guarantees. Not only do we establish this connection explicitly for the first time, but we also highlight a challenge in and propose a solution to the evaluation of selective prediction under DP. Further, we provide a comprehensive set of experimental results across common datasets quantifying (i) the exact drop in performance; as well as (ii) the loss in coverage needed to regain non-private utility. We believe that this work can help establish a bridge between the fields of selective classification and privacy with the goal of better understanding the interplay between these important sub-disciplines of trustworthy ML.
>
> > The necessity for new evaluation metrics is also still unclear to me. [...] Why do existing evaluation metrics fail in the case of private selective classification? Is there additional significance to the empirical results beyond the extension of privacy/utility tradeoffs to the case of selective classification?
>
> The reviewer is right that the purpose of Equation 4 is not to align accuracy; the purpose of Equation 4 is to evaluate the performance of a given selective classifier $(f,g)$. However, as noted by previous works [1], a meaningful comparison between different selective classifiers using Equation 4 is only possible if the accuracy on the full test set is aligned across all selective classifiers. That way, the effect of accuracy is fixed and Equation 4 solely measures SC performance.
>
> In non-private models, accuracy alignment is achieved via early stopping. Concretely, all selective classifiers are accuracy-matched to the worst performing model and models that could achieve a better utility level are stopped early in training. However, this approach does not carry over to private models. Private models trained using DP-SGD are iteratively trained to exhaust a desired privacy budget at the end of training. Early stopping a private model yields a more private model than was in fact targeted by the optimization process. However, it is important to ensure a consistent privacy level at which we evaluate private models. Moreover, training private models at varying $\varepsilon$ levels leads to decreasing accuracy as $\varepsilon$ decreases.
>
> These challenges of properly evaluating private models necessitate a new performance metric for selective classification, which is a key contribution of this paper. Although we could compute Equation 4 for all of our experiments on private models, it would be impossible to disentangle the effect of a drop in accuracy from a drop in SC performance; the score would trivially show a decrease in performance as a result of a loss in model utility due to privacy. __But just because model A is less accurate overall than model B, that does not necessarily mean that the model is less capable of identifying which examples it knows to classify well and which examples it doesn’t know how to classify well.__ We have made sure to strengthen this intuition throughout the paper.
>
> The proposed metric in Equation 6 circumvents this problem by computing closeness to an upper bound on the SC performance. This upper bound measures the optimal SC performance for a model with the particular full-coverage accuracy constraint of the underlying model. Post-hoc accuracy alignment is not required here as the bound is individual to the model and dependent on the overall model accuracy. Finally, not only does this metric allow us to quantify selective classification performance under DP, but it also allows for the comparison of non-private models across full-coverage accuracy levels. For instance, models with different accuracies might arise as a consequence of restrictions to the hypothesis class. As such, this metric can be also applied in more general selective prediction works that do not consider privacy.
>
> [1]: Feng, Leo, et al. "Towards Better Selective Classification." The Eleventh International Conference on Learning Representations. 2023.
> Questions:
>
> > In Figure 1, it would be nice to have the meaning of the various colors used explicitly stated, ideally in the caption. It's inferable from context, but additional description would probably save the reader time.
>
> Thanks for pointing this out! We have added the following sentence to the caption to increase clarity:
> The shaded color regions indicate degrees of confidence in the red (majority) or blue (minority) class, respectively. Stronger colors correspond to more confident predictions.
>
> > I also found the results in figure 4 and 5 were challenging to parse, mostly the significance of the blue and orange lines. Additional explanatory text to help the reader interpret these lines would be greatly appreciated.
>
> We have improved the caption of Figure 4 to clarify the role of the orange and blue lines in particular. All vertical lines shown in the plots constitute samples from the minority class. If a line is shown in blue, then it means that it is classified correctly, if it is shown in orange then it is classified incorrectly. The exact position of the vertical line on the x axis signifies the coverage level at which the minority point is accepted. This plotting technique allows us to inspect the behavior of selective classification for every point from the minority class and derive general distribution patterns over the coverage spectrum.
>
> ---
> **We hope that we have addressed the reviewer’s concerns and that the reviewer considers raising their score as a result.**

---

> > ### Author Response · Authors · 2023-08-18
> > **Thank you**
> >
> > We were happy to see that our rebuttal has helped the reviewer to understand the motivation underlying our submission better! We also thank the reviewer for raising their score and kindly ask the reviewer to consider updating their sub-scores for **Presentation** and **Contribution** as well.
> >
> > Unfortunately, although an interesting problem, the interplay of cryptographic adversaries and selective classification is beyond the scope of our work. We are not aware of this connection having been explicitly established so far. In terms of related work, we note that the connection between some selective classifiers, in particular Bayesian classifiers, and mutual information classifiers has been preliminarily explored [1]. Moreover, there exists a definition of DP that is based on a mutual information constrain which yields a definition of DP that sits in between $\varepsilon$-DP and $(\varepsilon, \delta)$-DP in terms of the provided guarantees and attack surface [2]. Finally, we do believe that quantifying predictive uncertainty and abstaining from prediction under high uncertainty does indeed constitute a valuable tool that can be of use in the bit security setup as described in the paper provided by the reviewer. As mutual information is intimately connected to entropy, it can quantify certainty in a prediction which can constitute a valuable signal for rejection. As such, future work should investigate to what extent recent advances in selective prediction can be helpful in the context of cryptographic adversarial applications.
> >
> > [1]: Hu, Bao-Gang. "What are the differences between Bayesian classifiers and mutual-information classifiers?." IEEE transactions on neural networks and learning systems 25.2 (2013): 249-264.
> >
> > [2]: Cuff, Paul, and Lanqing Yu. "Differential privacy as a mutual information constraint." Proceedings of the 2016 ACM SIGSAC Conference on Computer and Communications Security. 2016.

---

> > ### Comment · Reviewer_sqFp · 2023-08-18
> > **Thank you for the rebuttal!**
> >
> > [copying my previous comment to the appropriate rebuttal, apologies for any confusion]
> >
> > Thanks to the authors for help in understanding the challenges in SC evaluation in the private setting. I believe I misunderstood the existing approaches for evaluation, but the explanation and reference to prior work were enlightening. I've updated my score accordingly.
> >
> > This is unlikely to be useful, but I wonder what connections there might be between evaluating selective classifiers (in the non-private setting) and cryptographic adversaries. For instance, https://eprint.iacr.org/2018/077.pdf proposes a metric for evaluation of abstaining adversaries that follows from some analysis of the mutual information between the adversary's predictions and the correct label.

---

### Official Review · Reviewer_9iMW · 2023-07-09

**Soundness:** 4 excellent
**Presentation:** 4 excellent
**Contribution:** 4 excellent
**Rating:** 7
**Confidence:** 4

**Summary:**

The paper studies selective classification with differential privacy (DP). In particular, the authors conduct a comprehensive empirical study of different selective classifiers under DP. The key findings and contributions are:

* An existing SC method that uses checkpoints is suitable under DP.
* A novel accuracy-dependent selective classification score proposed by the authors helps compare selective classification performance across DP levels without explicit accuracy alignment.
* Selective classification performance degrades with stronger privacy.
* Recovering utility can come at a considerable coverage cost under strong privacy requirements.


**Strengths:**

* The paper initializes the study of the interplay between DP and SC. This is the first study of its kind. The proposed metric is effective in evaluating the performance of SC under different coverage. Thus, the significance of the proposed metric is beyond the study of the paper.
* The paper might have an impact on the community in the long run since it bridges previously disconnected fields (SC and DP). Thus, The paper pushes the community in a new, interesting/important direction.
* The paper is well-written and clearly organized. The evaluation is also comprehensive.

**Weaknesses:**

I do not find any obvious weaknesses in the paper. One minor point that could be improved is the explanation of the proposed metrics (Definition 3.1) in lines 255-260. It takes me some time to digest why the authors define the upper bound in this way.

Going beyond this, I think a more rigorous theoretical analysis and characterization between the trade-offs of SC and DP could be an interesting direction.

**Questions:**

Please improve the explanation of the proposed metrics (Definition 3.1).

**Limitations:**

The authors have adequately addressed the limitations.

---

> ### Author Rebuttal · Authors · 2023-08-06
>
> We thank the reviewer for their strong positive assessment of our work! We provide responses inline below.
>
> > One minor point that could be improved is the explanation of the proposed metrics (Definition 3.1) in lines 255-260. It takes me some time to digest why the authors define the upper bound in this way.
>
> We provide some more intuition for the bound: A perfect selective classifier accepts all correct points first and all incorrect points last. That is, for as long as possible, we want to maintain a perfect accuracy level of 100% as we accept an increasing number of points. We can only maintain perfect accuracy for as long as there still exists a data point that is classified correctly by the underlying model. This stops being the case once we have reached a coverage of $a_\text{full}$, where $a_\text{full}$ corresponds to the full-coverage accuracy. This is since an accuracy of $a_\text{full}$ means that a fraction $a_\text{full}$ of predictions are correct and a fraction $1 - a_\text{full}$ of predictions are incorrect. In the remaining part of the coverage spectrum, i.e. for coverage larger than $a_\text{full}$, the selective classifier now needs to accept all the incorrect points, which leads to a strictly monotonic decrease in accuracy. In particular, the full-coverage accuracy needs to be spread equally over the remaining accuracy spectrum which leads to $\frac{a_\text{full}}{c}$ as long as $a_\text{full} < c \leq 1$. We have updated the relevant lines in the paper to include this additional intuition for the coverage region larger than $a_\text{full}$ and hope that this increases clarity.
>
> We would further like to point out to the reviewer that Appendix B.3 (Figure 6 in particular) contains an upper bound reachability experiment. This experiment demonstrates that an optimal selective classification mechanism (i.e. a score that has a perfect acceptance ordering of test points) matches the bound exactly, showing that our formalism is indeed correct.
>
> > Going beyond this, I think a more rigorous theoretical analysis and characterization between the trade-offs of SC and DP could be an interesting direction.
>
> We agree with the reviewer that a more thorough theoretical treatment of the connection between selective prediction and differential privacy would be an exciting direction to pursue as part of future work. In particular, as pointed out to _reviewer atra_ below, we imagine that connecting selective classification and differential privacy via the membership inference angle could be a promising direction.

---

### Author Rebuttal · Authors · 2023-08-06

We thank all reviewers for their assessment of our work and have performed a few changes to improve our paper as a result of their feedback.

First and foremost, we were happy to see that **reviewers found our paper well written, that our idea of examining the connection between differential privacy and selective classification is novel and interesting, and that our experimental suite is comprehensive**.

Among other points, we have addressed the following key points of criticism in particular:

- **Relevance of a new performance metric for SC under DP**: We provide a more detailed reasoning as to why a new performance metric for SC is needed under DP. As accuracy alignment is not possible across models with various privacy budgets, established methods that rely on accuracy alignment fail. Our newly proposed metric computes the distance of the achieved accuracy-coverage tradeoff to an upper bound on the attainable SC performance (which we also propose in this work).
- **Additional results on partitioned deep ensembles**: We show that deep ensembles trained on partitioned data are inferior to deep ensembles trained on a single dataset but with DP composition.
- **Reasoning behind the grouping of SC methods**: We explain the grouping into post-processing and composition approaches in more detail, highlighting that taking multiple passess over the data is the decisive factor for a method to fall into the composition category.
- **Reliance on DP-SGD vs other ways of ensuring DP**: We remark that DP-SGD is the most widely used approach to ensure DP in deep neural nets and that alternate methods based on the functional mechanism or on output perturbation are not applicable in our models due to convexity assumptions.

We are happy to further engage with reviewers as part of the discussion phase and hope that the reviewers consider raising their scores.

---

### Decision · Program_Chairs · 2023-09-21

**Decision:**

Accept (poster)

**Comment:**

This work is the first to investigate the interplay between selective classification and differential privacy. Reviewers agreed that this is an interesting topic, and the experiments are thorough and convincing. Very few strong weaknesses were highlighted. I tend to agree overall, and recommend the paper for acceptance.